# Similar connectivity of gut microbiota and brain activity networks is mediated by animal protein and lipid intake in children from a Mexican indigenous population

**Elvia Ramírez-Carrillo**[1,2☯]*, **Isaac G-Santoyo**[1,3☯]*, **Oliver López-Corona**[4,5☯]*, **Olga A. Rojas-Ramos**[1,6], **Luisa I. Falcón**[7], **Osiris Gaona**[7], **Rosa María de la Fuente Rodríguez**[1], **Ariatna Hernández Castillo**[1], **Daniel Cerqueda-García**[8], **Andrés Sánchez-Quinto**[7], **Diego Hernández-Muciño**[9], **Javier Nieto**[10]

1 NeuroEcology Lab, Department of Psychology, UNAM, CDMX, México, 2 Investigadoras por México, Posdoc-CONACyT, Facultad de Psicología, Universidad Nacional Autónoma de México (UNAM), CDMX, México, 3 Unidad de Investigación en Psicobiología y Neurociencias, Department of Psychology, Universidad Nacional Autónoma de México (UNAM), CDMX, México, 4 Cátedras CONACyT, Instituto de Investigaciones en Matemáticas Aplicadas y en Sistemas (IIMAS), Universidad Nacional Autónoma de México (UNAM), CDMX, México, 5 Centro de Ciencias de la Complejidad (C3), Universidad Nacional Autónoma de México, CDMX, México, 6 Coordinación de Psciobiología y Neurociencias, Facultad de Psicología, Universidad Nacional Autónoma de México (UNAM), CDMX, México, 7 Laboratorio de Ecología Bacteriana, Instituto de Ecología, Universidad Nacional Autónoma de México, UNAM, Parque Científico y Tecnológico de Yucatán, Mérida, México, 8 Consorcio de Investigación del Golfo de México (CIGoM), Centro de Investigación y de Estudios Avanzados del Instituto Politécnico Nacional, Unidad Mérida, Departamento de Recursos del Mar, Mérida, Yucatán, México, 9 Laboratorio de Agroecología Instituto de Investigaciones en Ecosistema y Sustentabilidad, UNAM, Morelia, México, 10 Laboratorio de Aprendizaje y Adaptación, Facultad de Psicología, Universidad Nacional Autónoma de México (UNAM), CDMX, México

☯ These authors contributed equally to this work.
* isantoyo@psicologia.unam.mx (IGS); elviarc@comunidad.unam.mx (ERC); lopezoliverx@gmail.com (OLC)

**Data Availability Statement:** All sequences obtained were uploaded to the NCBI database

## Abstract

The gut microbiota-brain axis is a complex communication network essential for host health. Any long-term disruption can affect higher cognitive functions, or it may even result in several chronic neurological diseases. The type and diversity of nutrients an individual consumes are essential for developing the gut microbiota (GM) and the brain. Hence, dietary patterns might influence networks communication of this axis, especially at the age that both systems go through maturation processes. By implementing Mutual Information and Minimum Spanning Tree (MST); we proposed a novel combination of Machine Learning and Network Theory techniques to study the effect of animal protein and lipid intake on the connectivity of GM and brain cortex activity (BCA) networks in children from 5-to 10 years old from an indigenous community in the southwest of México. Socio-ecological conditions in this nonwestern lifestyle community are very homogeneous among its inhabitants but it shows high individual heterogeneity in the consumption of animal products. Results suggest that MST, the critical backbone of information flow, diminishes under low protein and lipid intake. So, under these nonwestern regimens, deficient animal protein and lipid consumption diets may significantly affect the GM-BCA connectivity in crucial development stages.

under the Bioproject number PRJNA593240 with the link: https://www.ncbi.nlm.nih.gov/bioproject/?term=PRJNA593240.

**Funding:** This project was funded by UNAM-PAPIIT [grant numbers IA209416, IA207019], DGAPA postdoctoral Fellowship, CONACYT Ciencia Básica [grant number 241744] and Instituto de Ecología-UNAM [L.I.F].

**Competing interests:** The authors have declared that no competing interests exist.

Finally, MST offers us a metric that unifies biological systems of different nature to evaluate the change in their complexity in the face of environmental pressures or disturbances. Effect of Diet on gut microbiota and brain networks connectivity.

## Introduction

The gut microbiota-brain axis is a complex system that integrates endocrine, immunological, and neuronal signals between symbiotic microorganisms, the gut, and the brain [1]. Here, the host's diet can be a fundamental factor for the functioning of this axis, especially during critical developmental windows for the gut microbiota (GM) and the brain [2, 3]. Maturation of both systems occurs in parallel, and childhood is one of the most dynamic periods of change. Thus, the GM interaction with the host diet during such critical periods has the potential to alter brain-gut signaling profoundly, affect health throughout life, and even increase the risk of neurodevelopmental disorders. For instance, during this age, the ecological dynamic of bacteria that compose the host' GM reaches a maturity that resembles an adult GM, incrementing total diversity and abundance and establishing strong symbiotic mutualistic relations with their hosts. For the brain, it causes the elimination of the extra synapses, decreasing cortical gray matter levels, and the formation of new neuronal connections producing a phase of high plasticity throughout much of the brain [4]. On the other hand, enhanced nutrition incorporating nutrients of animal origin, such as protein and lipids, during these windows of neurodevelopment might have a lasting effect on behavior and cognition in adulthood. For instance, increasing animal protein intake during the first week after birth may protect preterm children's brain structural and functional development. In the same way, serial Magnetic Resonance Images from birth show that animal lipid and protein intake during the first few weeks positively predicted larger brain volumes and cognitive outcomes at 18 months [5]. Moreover, certain dietary deficiencies present in animal products during the first two years of life, such as iodine or iron, create adverse cognitive effects that are not reversed by a later adequate diet [6]. Additionally, animal protein-energy malnutrition is responsible for more specific damage to the hippocampus and cortex [7]. Animal lipids, in turn, are essential for synthesizing neuronal membranes, in the construction of several neural structural elements such as myelin, in the production of signaling components, and are solvents of a wide variety of non-polar extracellular and poorly soluble cellular constituents [8]. Beyond these aged-related effects, the two nutrients consumption continue to influence cognition by acting on molecular systems or cellular processes vital for maintaining brain functioning [9].

Similarly, GM is also affected by the host's consumption of these two nutrients, influencing their taxonomic properties in terms of diversity, abundance, and presence [10, 11]. Nevertheless, the impact of diet or any host ecological pressure on a complex and interactive system like the GM and brain needs to go beyond knowing what bacterial groups are involved [12]. It is not enough to characterize the elements that constitute the systems but also their interactions. Consequently, it requires an ecological approach that considers these interactions in an explicit manner, and Network Theory usually makes this approximation. Here, it is possible to obtain valuable insights about functioning by analyzing the structure of the networks, for example, its connectivity, that ultimately will mediate the way information structure itself, under the interactions coded in the network. For example, in a recent work, it has been suggested that certain behavioral disorders, such as the development and persistence of depression, may be associated with a complex network of communication between macro-and micro-organisms capable of modifying different host's neurophysiological components [13]. In this sense, to understand

the parallelism between brain and GM development and how these biological systems can be affected by environmental pressures imposed by the host (i.e., diet), it is necessary to determine patterns of connectivity and complexity between these systems. For these reasons, in this work, we show a case study that evaluates how animal protein and lipid consumption may impact the bacteria GM and brain cortex connectivity in childhood. To accomplish this, we implemented a novel combination of Machine Learning and Network Theory techniques, which allow us to measure and unify the complexity of two biological systems of different natures; bacteria and neurons.

On the other hand, in terms of this complex systems approach, Network Theory Methods (NTM; one of the main streams in complexity) have become more widespread in neurobiological studies, for example, to determine brain dysfunction mechanisms [14–18]. Nevertheless, several network metrics, like node centrality indices, assume different significance on a local or global scale [19, 20]. This discrepancy could be a limitation in measuring complex systems because a good complexity metric needs to be scale invariant. In that respect, Saba and Coworkers [21] advocated using the Minimum Spanning Tree (MST) as a straightforward solution, ensuring accuracy, robustness, and reproducibility that avoid methodological confounding processes that may occur in other NTM.

These advantages are in part due to the way MST takes both topological properties and functional connectivity information into account [22–24]. MST is one of the best-studied optimization problems in computer science. The task is to calculate a tree that connects all the vertices of a network (a spanning tree) using only edges so that the total weight is minimum among all possible spanning trees [25]. This kind of Network Theory analysis has been getting much attention recently because it opens a multidisciplinary approach that allows us to analyze several biological components, such as the brain and the GM, simultaneously as complex systems in a straightforward computable way [26], and not merely as a metaphor.

Therefore, we will follow this complex system approach to study whether protein and lipid consumption in children can influence the connectivity of both brain cortex activity (BCA) measured by quantitative electroencephalography (qEEG) and GM determined by 16S rRNA gene high-throughput sequencing. To accomplish this, we have selected an indigenous Mexican community belonging to a pre-Hispanic ethnic group, self-appointed Me'phaa. The Me'phaa people inhabit a small region named "La Montaña Alta" in Guerrero, México. It is one of the more contrasting groups in terms of lifestyle compared with typical westernized urbanized areas [27–29]. Due to geographical isolation, social customs, and poverty, subsistence farming provides most of their diet. For example, the recollection of wild edible plants; consumption of some fruits and vegetables cultivated in garden plots; meanwhile, animal-based products are available almost by hunting, consumed during special occasions, and are not part of the daily diet [30]. Hence, these animal nutrients' consumption is highly limited in this community but shows a relevant inter-individual variation associated with familiar acquisitive capacity [30] Interestingly, this community shows close inter-individual similarities in other factors that may influence BCA and GM, such as the use of allopathic medication, access to health services, economic status, parent and child scholarly, and access to computational and communications technologies [10, 31]. Consequently, this human population offers a unique model that naturally homogenizes these factors in a proper ecological way. Consequently, this human population offers us a unique model that controls these factors in a natural and ecologically valid way. We hypothesize that inter-individual variations in the access to these limiting nutrients would affect children's GM and BCA networks. Connectivity in GM and BCA networks evaluated by MST would be lower in children with less consumption of animal proteins and lipids.

## Materials and methods

### Procedures to recruit participants

All procedures for testing, experimental protocols and recruitment were approved by National Autonomous University of Mexico Committee on Research Ethics (FPSI/CE/01/2016), and run in accordance with the ethical principles and guidelines of the Official Mexican Law (NOM-012-SSA3–2012). All parents or caregivers of children participants signed an informed consent. Participant recruitment was done through the Xuajin Me'Phaa non-governmental organization, which is dedicated to the social, environmental and economic development for the indigenous communities of the region (see video from this organization [32]). Xuajin Me'Phaa has extensive experience in community-based fieldwork and has built a close working relationship with the community authorities. The trust and familiarity with the community customs and protocols have previously led to successful academic collaborations (i.e. [33–35]). Therefore, Xuajin Me'Phaa served as a liaison between the our research group and the communities, offering mainly two important factors in data collection: the informed consent of community members and participants, and two trained interpreters of Me'Phaa and Spanish language of both sexes. The whole procedure was carried out within their own communities.

### Study site and participants

The present study is composed by a sample size of 31 indigenous children between 5–10 years old (12 males and 19 females) who inhabits in two small Me'phaa Communities from "La Montaña Alta" from the state of Guerrero; Plan de Gatica ($17_7049.5552"N99.70,EASL510m$) and El Naranjo ($17_9054.0036"N98570,50.9832x201D;W,EASL860m$). The distance between the two indigenous communities is almost 30 km, and the socioeconomic and cultural patterns are similar [30, 33]. In order to evaluate sexual differences in participant's body composition, we obtained standardized basic anthropometric measurements (Table 1). These included weight (i.e. kg) using a bioimpedance digital scale (OMROM-HBF-514C), head and brachial circumference using an ergonomic circumference measuring tape measure (SecaTM 201), and height using a portable leveled stadiometer (HM200P, PortStad) placed on the wall at 2m height.

### Protein and lipid intake estimation and categorization

To determine the daily frequency of protein and lipid consumption in children participants, we adapted the food frequency questionnaire (FFQ) from the Mexican national health system ENSANUT-2012 (https://ensanut.insp.mx/) with the available food in Me'phaa communities. The ENSANUT uses a probabilistic multistage stratified cluster sampling design to represent different regions of the country. Briefly, every child's mother (or caregiver)answered the FFQ

**Table 1. Anthropometric measurements of study participants, reported as mean +/- standard deviation.** Using a Wilcoxon-Mann–Whitney test we found no significantly difference for any variable in terms of gender.

|  | Males mean | Females |
| --- | --- | --- |
| Age | 8.08 +/- 1.68 | 7.89 +/- 1.63 |
| Height (cm) | 118.67 +/- 10.22 | 117.37 +/- 13.09 |
| Weight (Kg) | 22.42 +/- 4.05 | 22.55 +/- 8.38 |
| Head circumference (cm) | 50.73 +/- 1.49 | 50.22 +/- 1.82 |
| Brachial Perimeter (cm) | 17.78 +/- 1.44 | 16.95 +/- 1.47 |
| Protein intake/month (gr/month) | 279.46 +/- 68.20 | 281.99 +/- 136.20 |
| Lipid intake/month (gr/month) | 215.47 +/- 50.78 | 210.39 +/- 90.54 |

registering the composition of the child's diet by counting the number of times the food is consumed per day and week (see Questionaries in S3 File). From this frequency, we calculated an approximate animal lipid and protein consumption in grams per month (i.e., 30 days). We approximated this by summing the monthly reported portions and the grams of protein and lipids each food contributes, according to the Mexican System of Equivalent Foods [36]. Following the record of consumption with international standard, we considered milk, yogurt, pork, beef, chicken, egg, butter, and fish as the main food that contributed to these two macronutrients (S3 File).

In order to compare the connectivity of BCA and GM networks of children (n = 31) with different rates of animal protein and lipid consumption, for subsequent analysis, we divided the daily frequency of these two macronutrients into two categories; Low (L) or High (H) consumption. To accomplish this, we used the mean intake of the total sample to categorize participants into one of these two groups. The L group comprised participants with consumption below the mean obtained, while the H group comprised participants with consumption above the same mean. For the case of protein, we obtained 11 participants in H (mean = 407.48 gr/month, SD = 67.58 gr/month) and 20 participants in L (mean = 211.94 gr/month, SD = 59.25 gr/month), while for lipids we obtained 9 participants in H (mean = 278.22 gr/month, SD = 74.80 gr/month), and 22 in L (mean = 185.41 gr/month, SD = 60.28 gr/month). Using the mean of the total sample to categorize the participants allowed us to obtain two groups with significantly different consumption values for both nutrients, evaluated through a T-test for independent samples between the High vs. Low categories. In both nutrients, the differences were statistically significant (Protein: t = 8.065, p < 0.001; Lipid: t = -3–30, p < 0.001; Fig 1).

There are 11 individuals with high protein intake, 9 with high lipid intake, and 6 individuals that are both in the high protein and lipid intake, corresponding to 42.8% shared. On the other hand, there are 20 individuals with low protein intake, 22 with low protein intake, and 16 both with low protein and lipid intake, corresponding to 61.5% shared.

Fig 1, Boxplot of nutrient intake for high and low consumption.

## Brain cortex activity

We determined BCA in children participants in terms of absolute power obtained through qEEG during a resting-state period. We used a GRAEL 4K EEG Amplified to recorder the brain cortex electrical signals, electrode caps, and the signal acquisition program Profusion EEG5. For the acquisition process, we used small (50–54 cm) and medium (54–58 cm) size Electrocap universal caps at 19 localizations according to the 10–20 International System [37]. The caps were chosen depending on the cephalic circumference. We acquired a resting-state EEG record through a monopolar montage using a reference for each ipsilateral ear lobe and an extra electrode at the left eye's external canthus to record eye movements.

## Signal acquisition and qEEG processing

Data were recorded with a bandwidth of 1 Hz to 70 Hz, an on-line Notch filter, and a sampling rate of 512 Hz; impedances below 10 kΩ were maintained in all the electrodes; a 5-minute open-eye and 5-minute closed-eye resting-state recording were performed for each participant. We record brain activity using a counterbalance between individuals, starting with eyes open or eyes closed depending on their disposition. The record lasts 15–20 minutes per individual, switching conditions to achieve two uninterrupted minutes of each condition (eyes open and eyes closed). An offline visual inspection was performed in the Profusion EEG5 to obtain 2 minutes running without each condition's artifacts. Data were analyzed using a personalized script in Matlab and the Fieldtrip toolbox [38]. We applied filters above 1 Hz and

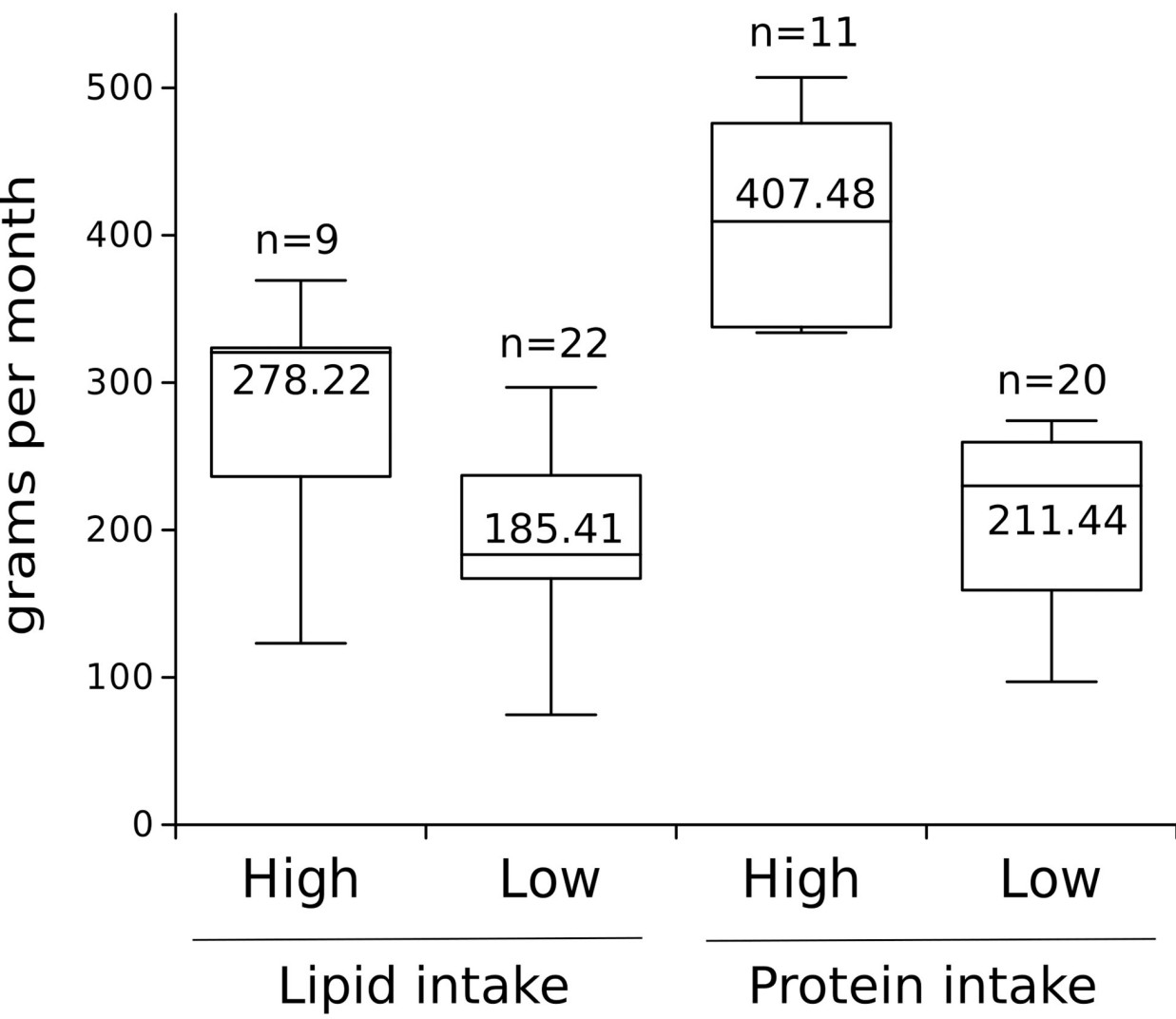

**Fig 1. Boxplot shows the intake values including number of participants for each case (above the box), and median values (inside it).** Both nutrients categories are significantly different (Lipid: t = 3.30, $p < 0.001$; Protein t = 8.065, $p < 0.001$).

below 40 Hz. A variance-based distribution method between trials and a second visual inspection was performed to obtain 39 to 42-second artifact-free epochs of each subject and each condition. Finally, the data were transformed to a logarithmic scale to proceed with the power analysis. We obtained the absolute power using a Fourier Analysis for each derivation, participant, and condition, by classic broad bands Delta (1–3 Hz), Theta (4–7 Hz), Alpha (8–14 Hz), and Beta (15–30 Hz). We finally created five brain nodes or regions of interest (ROI's) of general brain activity; we averaged the absolute power from three electrodes per ROIS's. ROIS's contained the following electrodes; left anterior (Fp1, F3, F7), right anterior (Fp2, F4, F8), left posterior (P3, T7, O1), right posterior (P4, T8, O2), and midline (Fz, Cz, Pz).

### GM taxonomic assignation

Taxonomic assignation of GM bacteria in each participant was obtained from fecal purified 16S rRNA fragments (20 ng per sample) sequenced on an IlluminaMiSeq platform (Yale

Center for Genome Analysis, New Haven, CT, USA). All sequence data used in the present study have been previously reported in [33]. Briefly, the sequenced reads (Bioproject number PRJNA593240) were denoised with the DADA2 plugin to resolve the amplicon sequence variants (ASVs) [39]. Both forward- and reverse-reads were truncated at 200 pb, and chimeric sequences were removed using the "consensus" method. Representative ASVs sequences were taxonomically assigned using the "classify consensus-vsearch" plug in [40], using the SILVA 132 database as a reference [41]. An alignment was performed with the MAFFT algorithm [42]. After masking positional conservations and gap filtering, a phylogeny was built with the FastTree algorithm [43]. The abundance table and phylogeny were exported to the R environment to perform the statistical analysis with the phyloseq [44] and ggplot2 packages. Plastidic ASVs were filtered out of the samples, which were rarefied to a minimum sequencing effort of 21,000 reads per sample. Alpha or total diversity was calculated through Shannon's Diversity Index and Observed ASVs. We performed a Welch two-sample t-test to determine differences in total diversity through protein and lipid consumption levels. We estimated Beta diversity (or degree of community differentiation, in relation to protein and lipid consumption) by computing unweighted UniFrac distances, obtaining statistical differences by a permutational multivariate analysis of variance using distance matrices (PERMANOVA). Additionally, we performed a differential abundance analysis with the DESeq2 library [45] to determine the main discriminant ASVs by the levels of protein and lipid consumption. ASVs considered as discriminants were those that statistically differ between the level of lipid or protein consumption at $p < 0.01$, corrected with the false discovery rate (FDR) method.

## Network analysis for EEG and GM data

Following the work done by Saba and co-workers [21], we took the five BCA nodes arrangement (front left, front right; posterior left, posterior right; and midline) to calculate a Mutual Information (MI) matrix using the mpmi R Package [46]. Then, we constructed the corresponding networks in Cytoscape using the MI matrix coefficients as weights. Afterward, we calculate the Minimum Spanning Tree (MST) in Gephi [47]. In our context, given that each weight is a MI-connectivity value, the weight of the MST (the sum of the MST's weights) is a measure of both structural and functional brain connectivity [48]. In network theory, a tree is an undirected network in which any two vertices are connected by exactly one path, or equivalently a connected acyclic undirected graph. If a tree includes all vertices then it is a spanning tree, and if that spanning tree satisfies the condition of giving the minimum possible total edge weight, then this spanning tree is named as the MST. So, the MST finds an edge of the least possible weight that connects any two trees in the forest. This algorithm finds a subset of the edges that forms a tree that includes every vertex, where the total weight of all the edges in the tree is minimized.

Fig 2 shows a basic flow diagram for data analysis described in Methods.

For the case of GM, we carried out an initial exploratory data to identify whether the effect of different levels of protein and lipid consumption is expressed in a small number of ASVs, or if, on the contrary, the effect is systematically distributed in the network. For this, we used the Machine Learning algorithm of Random Forest [49] implemented in R under the randomForest package [50], and we identified essential ASVs (equivalent to BCA nodes) in terms of its Mean Decrease Gini value. This value tells us about the node's relative importance for the classification process in terms of high vs. low nutrient intake. Although we found that not all ASVs participate informationally in protein or lipid intake differences (Fig 5a and 5b), classification analysis pointed to the fact that the way GM responds to protein and lipid intake

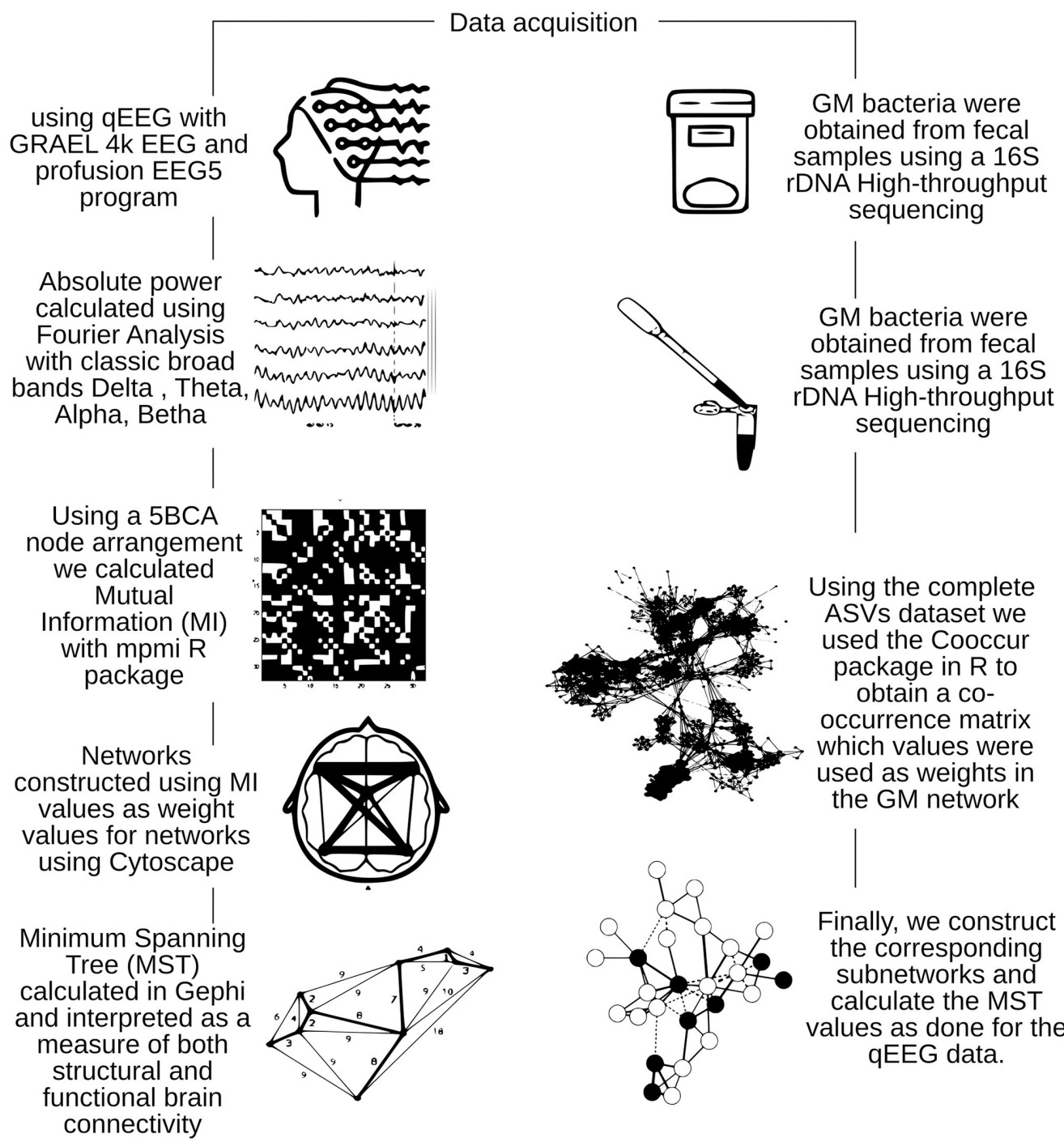

**Fig 2. Basic flow diagram for data analysis.**

differences is distributed among the network. For this reason, we used the complete ASVs data set as input to a co-occurrence analysis using Cooccur package in R [51]. The co-occurrence matrix obtained was then used as weights values in the GM network. Finally, we calculate MST values for GM as was done for the BCA data.

**Table 2. Values above gray boxes correspond to mean and standard deviation for each nutrient intake condition; Gray boxes shows U Mann-Whitney test.**

|  | PROTEIN INTEAKE | | | | LIPID INTAKE | | | |
|---|---|---|---|---|---|---|---|---|
|  | High | | Low | | High | | Low | |
| Height (cm) | 8.11 | 1.69 | 7.4 | 1.3 | 8.55 | 1.87 | 7.11 | 1.26 |
|  | *U = 82, p = 0.24* | | | | *U = 72, p = 0.23* | | | |
| Weight (kg) | 118.72 | 9.47 | 117.4 | 13.24 | 115.88 | 13.85 | 118.68 | 11.25 |
|  | *U = 100, p = 0.67* | | | | *U = 95, p = 0.86* | | | |
| Weight (kg) | 21.73 | 4.22 | 22.92 | 8.12 | 20.96 | 4.18 | 23.12 | 7.79 |
|  | *U = 107.5, p = 0.91* | | | | *U = 93, p = 0.79* | | | |
| Weight (kg) | 17.21 | 1.43 | 17.32 | 1.56 | 17.35 | 1.83 | 17.25 | 1.36 |
|  | *U = 89, p = 0.5* | | | | *U = 83, p = 0.61* | | | |

## Results

As for general information about the participants, in Table 1 we show their anthropometric measurements according to sex. Using a Wilcoxon-Mann–Whitney test we found no significantly difference for any variable in terms of sex, so in further analysis we did not divide data in function of it.

In Table 2 we now focus on the variable used for network theory analysis: protein and lipid intake. We show that there is no significant difference in terms of anthropometric measurements (Gray boxes; Wilcoxon-Mann-Whitney test) and the protein or lipid intake. Values above gray boxes correspond to mean and standard deviation for each nutrient intake condition.

### Brain cortex connectivity

Fig 3 shows the MI matrices of the BCA for all classic bands in each ROIs and resting state condition (i.e. Open and close eyes). Here, lines represent the level of MI in all possible ROIs combinations.

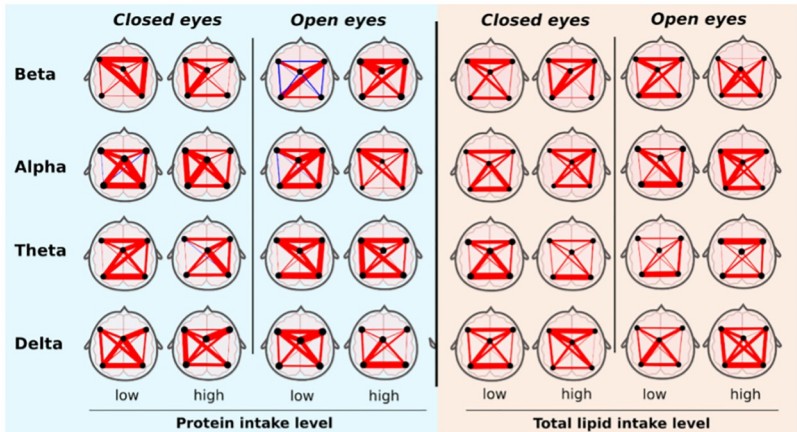

**Fig 3. We show a resting-state EEG Mutual Information networks (MI), considering five regions of interest (ROIs): Front left, front right; posterior left, posterior right; and midline.** MI was obtained for each broad brain band in the two resting-state conditions, Open and Closed eyes. Similar to what is occurring in a correlation analysis, the red color indicates positive relationship between ROIs and blue negative. The thickness of the line represents the intensity of the relationship.

Most of the MI networks show a positive relation (Fig 2). However, the magnitude of the relation through ROIs nodes in all broad bands was qualitatively different, depending on protein and lipid intake levels. Notably, we observed that low protein intake results in negative MI values between left, right, anterior and posterior nodes in the beta band during open eyes (i.e. blue lines), representing the condition of the highest cognitive demand, while high protein intake results in a strongly positive relation. On the other hand, we found that except for the beta band for close eye condition, MST weights built from MI networks showed a systematic increase in the highest protein group for all classic broadbands; hence, higher brain connectivity (Fig 4). We also observed that except for the beta band in open eyes and the delta band in closed eyes condition, the increment in MST weight was also observable for the highest lipid group. The increase ranges from 12%, as occurred in the Delta band in open eyes condition, to even more than 80% in the alpha band for closed eyes condition.

Fig 4, shows the value of minimum spanning tree (MST) of absolute brain power networks for protein or lipid intake level in both condition A) closed or B) Open eyes.

## GM connectivity

The GM composition and abundance was not affected by the level of protein or lipid intake. Both nutrients did not affect the GM total diversity nor Shannon index (Protein: p = 0.51;

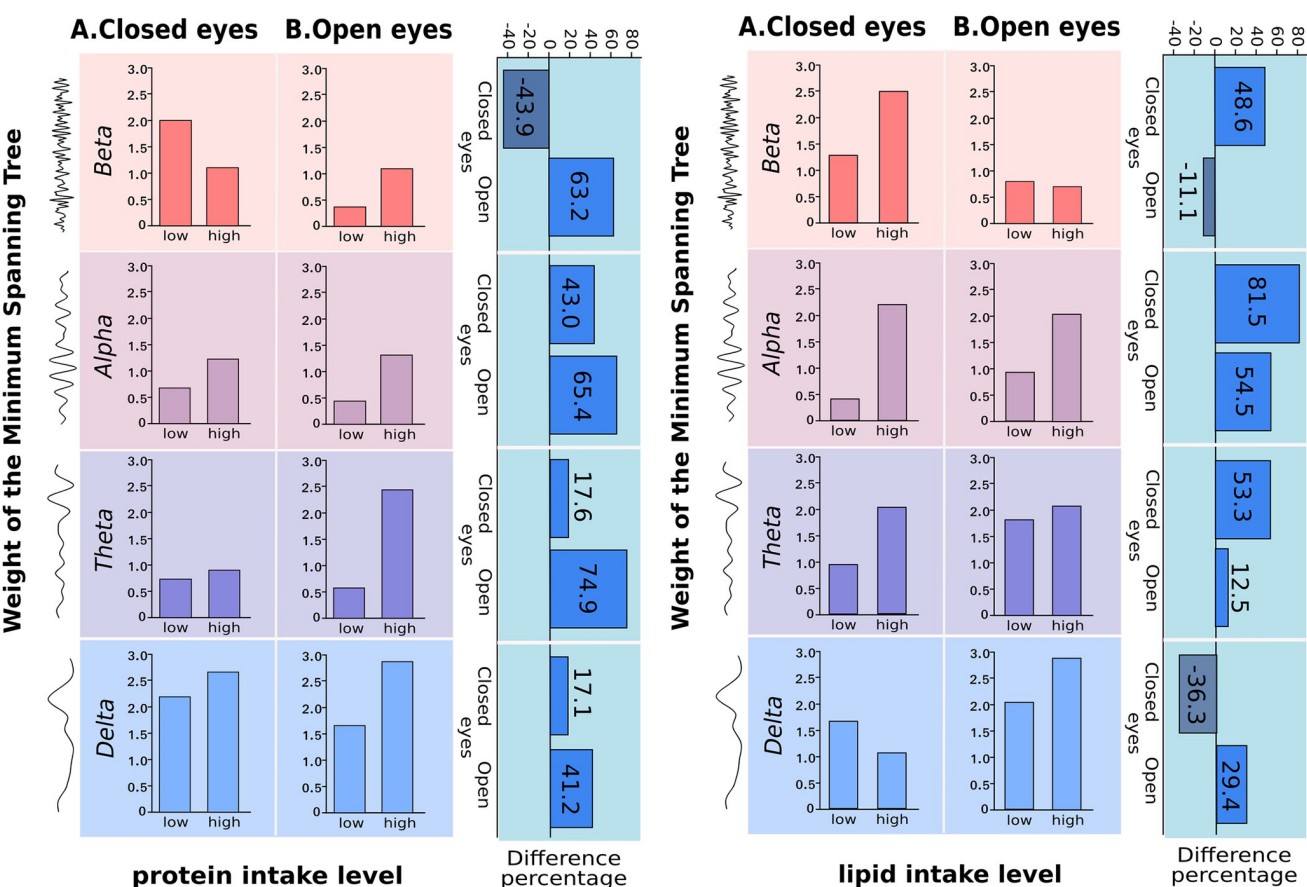

**Fig 4. Minimum spanning tree (MST) of absolute brain power networks for protein or lipid intake level in both condition A) closed or B) Open eyes.** Weight of the MST represents a measure of total structural and functional network connectivity (the higher the value, the higher connectivity). The complementary right-side subpanel (one for protein intake and one for lipid intake) shows the difference percentage for the condition, each value is annotated inside the corresponding bar of the sub-panel (light blue is used for positive difference, dark blue for negative difference).

Lipid: p = 0.98, S1a, S1c Fig in S1 File) nor Observed ASVs (Protein: p = 0.51; Lipid:0.43, S1a, S1c Fig in S1 File). Also, there were no discriminant or beta diversity differences, between high and low protein (PERMANOVA: F = 0.91, p = 0.65, S1C Fig in S1 File) or lipid consumption groups (PERMANOVA: F = 0.87, p = 0.76, S1D Fig in S1 File). Finally, two Faecalibacterium ASVs (ASV_1 and ASV_2) and one undetermined ASV were the only discriminant ASVs found between low and high groups in both macronutrients. Faecalibacterium ASV_1 was more represented in the high protein group but Faecalibacterium ASV_2 was for the high lipid group. The undetermined ASV was higher only for the low lipid group (S2a, S2b Fig in S1 File). Nevertheless, GM connectivity assessed through MST was different between intake conditions.

Fig 5 shows the Mean Decrease Gini value on a log-normal scale, representing the relative importance for every ASVs identified (i.e. approximation of species) for each nutrient. For protein, only 740 ASVs contribute informationally to the Random Forest (Fig 5a), while for the case of lipid, it turned out to be a similar number of ASVs, resulting in 721 (Fig 5b).

Similar to BCA, GM connectivity also diminished in the low protein and lipid intake groups. Fig 6 shows GM connectivity calculated as the total weight of the corresponding MST of low Vs. high intake of both macronutrients. Compared with high consumption, Low consumption of protein and lipid diminished the MST weight in 49.98% and 51.38% respectively (Fig 6).

## Discussion

Here we used novel methodologies [13, 21] to analyze the parallelism in connectivity between the BCA and GM networks. This ecological/complex approach allows us to understand how information flows are constrained by external conditions, like diet patterns, modifying the GM ecosystem network topology and BCA in a resting state. In this case study, we test how animal protein and lipid consumption influence the BCA and the GM networks' connectivity in childhood. The GM and the brain undergo a critical developmental window in this period of life, in which essential factors, such as diet, might profoundly influence the gut-brain signaling in adulthood [52]. Our results suggest that under this non-western diet, low animal protein and lipid intake may lead to non-neglecting effects on BCA and GM in terms of network connectivity. For example, brain connectivity measured by the MST (or the critical backbone of information flow in the network) improves for almost all brain broadbands under open eyes condition (cognitively demanding) when protein or lipid intake increases. Interestingly, optimal brain functioning for these ages may present recurrent activity in alpha rhythm, and the absence of the beta rhythm during closed-eyes [53]. Regarding brain connectivity, this phenotype was only observable in children with high protein and lipid consumption. Furthermore, the alpha rhythm was the most influenced by the protein and lipid consumption level, increasing brain connectivity on this oscillation when children consume higher amounts of these nutrients. This kind of oscillation is also essential to regulate the top-down functional inhibition of neuronal processing, representing a neurophysiological resource for optimal processing capabilities in the visual system, regulating the competition between target and distracting stimuli [54]. Complementary with alpha, the beta rhythm was another significant oscillation affected. In this case, beta oscillations are expected to have a greater frequency in open eyes, and the opposite is for closed eyes [55]. We found that brain connectivity in the beta rhythm follows this pattern but only for children with the highest protein consumption. On the other hand, BCA in the resting state is considered an indicator of cerebral self-modulation, a unique feature of the cerebral cortex that ensures that information flows across the entire neural network and flows to the right place at the right time [56]. Furthermore, this organized activity

## A. Protein random forest analysis

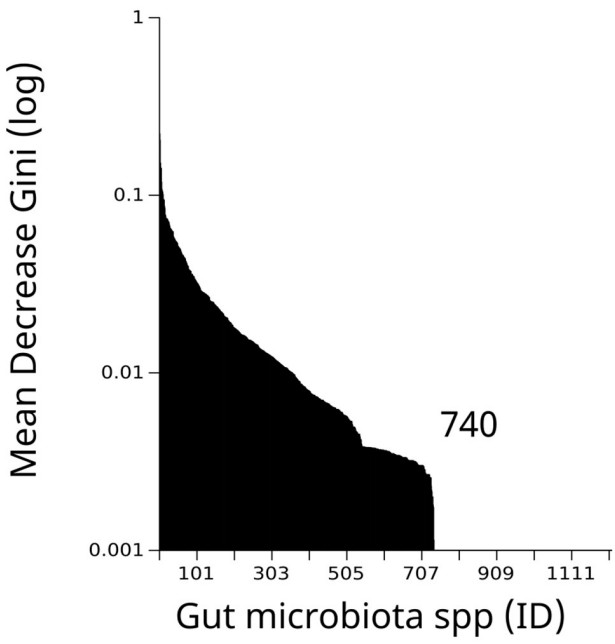

## B. Lipid random forest analysis

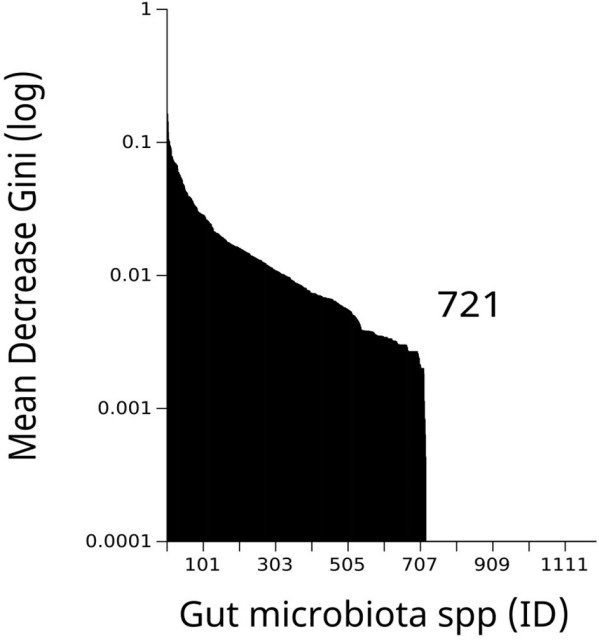

**Fig 5. Results from Random Forest of Mean Decrease Gini values that tell us the relative importance of different ASVs to the classification processes into low Vs high intake for both protein (gray) and lipid (black) intake.**

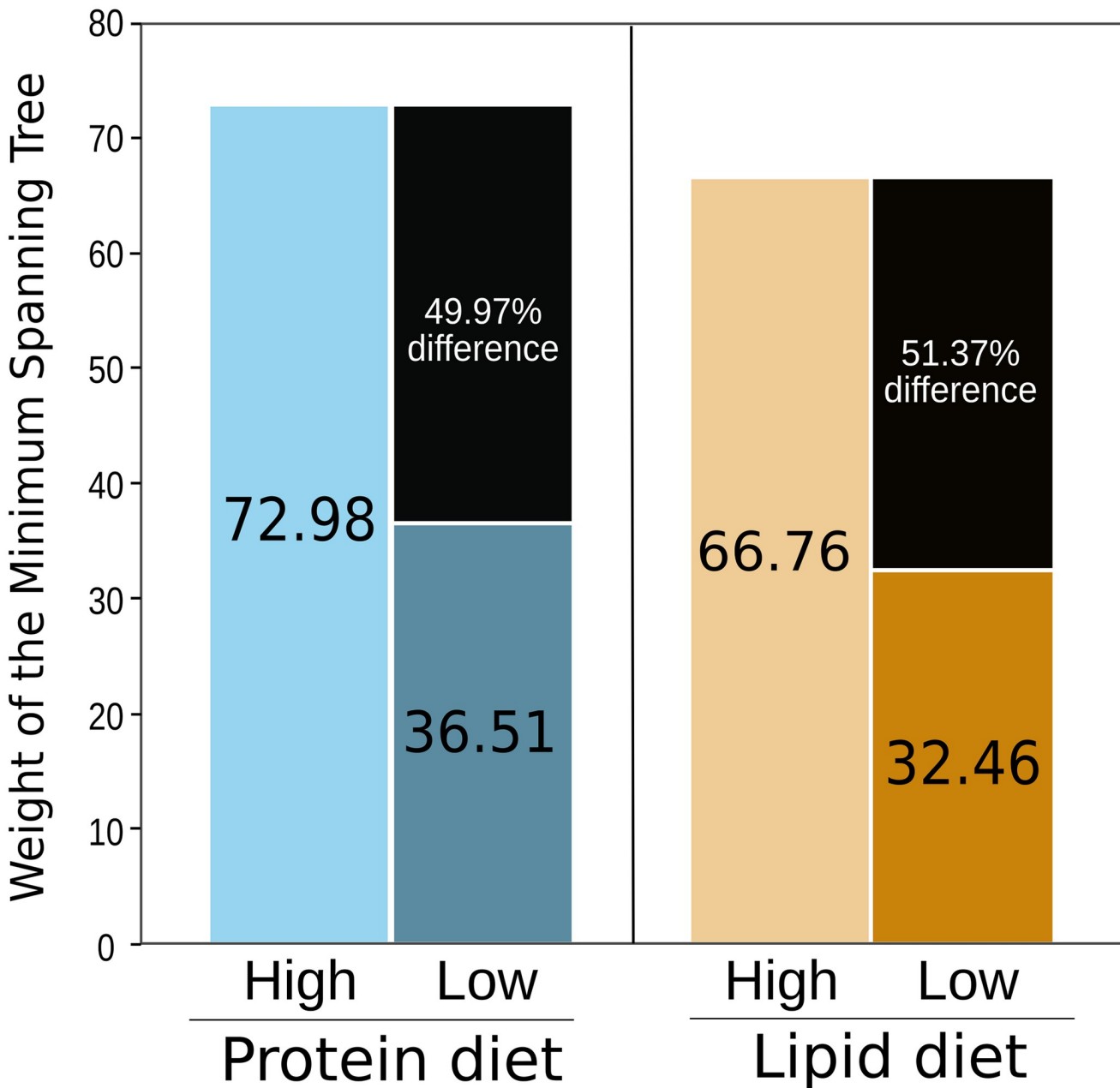

**Fig 6. Weight of the Minimum Spanning Tree (MST) as a measure of total structural and functional network connectivity (the higher the value, the higher connectivity) for both protein and lipid intake.** There is a notable difference percentage in the weight of MST of the high intake treatment compared with the low intake treatment.Light colors are used for high intake and dark colors for low intake, and the corresponding MST weight values are annotated inside each. Value of difference percentage is annotated in black sub-bar.

promotes an electrical coupling that establishes temporal windows in which neurons respond to certain stimuli facilitating or limiting the synaptic transmission for different cognitive processes [57, 58]. Nevertheless, given that we did not measure any of these cognitive functions, an essential prospect of this study is to evaluate whether network connectivity patterns found here are associated with demanding cognitive performance, such as working memory or executive functions. A novelty and valuable aspect of implementing MST measurements in

determining BCA are that if we consider the brain as an information transport network, where cerebral areas are nodes, and connections edges [26]. Then, MST enables the assessment of both structural (anatomical) and functional (the statistical relationship between two nodes) connectivity [58], and it might represent the critical backbone of information flow [21]. Additionally, measure like the MST considers, at the same time, the interaction of all the cortical components measured (i.e., brain cortex electrodes). Contrary to traditional measures of functional brain connectivity, which only represent a correlation matrix of absolute or relative power between two brain nodes [59]. This approximation could be a novel implementation for determining "complete" brain networks integrating all system components. For instance, MI the entropy-based metric we used to obtain MST, is vastly more potent than correlation; MI can detect nonlinearities, and it is additive and ergodic [60]. Besides, MI captures the brain network's physiologically relevant features better than correlation, as Zhang and co-workers [61] found in children with epilepsy, Sun and co-workers [62] within post-stroke patients with different levels of depression, or Gaubert and co-workers [63] as a helpful biomarker for preclinical Alzheimer's.

Concerning GM, we did not find differences in the total diversity (i.e. alpha diversity) or discriminant composition (i.e. beta-diversity) between high and low frequency of consumption in both macronutrients. We only found three discriminant ASVs; One for protein (Faecalibacterium ASV_1) and two for lipid (Faecalibacterium ASV_2 and an unassigned ASV). These results indicate that GM, in terms of composition and abundance, is very similar in these indigenous children and is not dependent on lipid or protein consumption. However, despite having a similar GM, we found relevant differences in connectivity. Here, we observed the same effect for the GM as for the BCA; diminishing their network connectivities under the low protein and lipid intake. Interestingly enough, from an ecological perspective, a significant level of connectivity dissipates the effect of perturbations in the distribution of species and enhances stability. Hence, as suggested by Equihua and co-workers [64], this would lead to a loss in GM resilience/antifragility (system capacity to respond to perturbations) even though the two groups show similar compositions and abundances. In consequence, under a low animal-nutrient diet (especially in childhood), GM, brain or both, could lead to a systemic loss of health, with potential long-term effects in the axis that these two components interact. Given these results, a logical subsequent goal is to compare another population with different patterns of protein and lipid consumption, for example, a westernized population. Nevertheless, this could arise several methodological challenges. This kind of population differs not only in the frequency of consumption of these nutrients, but considerable differences in subyacent factors that may influence BCA and GM, such as the use of allopathic medication, access to health services, economic status, parent and child scholarly and access to computational and communications technologies [10, 31]. Consequently, they could confuse the effect of protein and lipid intake alone. Also, a longitudinal study modifying the amount of protein and lipid consumed in the study population would be a natural extension for future work.

As we have discussed above, current research in the field is pointing to the importance of doing microbiota studies in nonwestern populations, and how minorities are usually underrepresented in mainstream literature. This is relevant not just because they represent very interesting contrasting populations, but also to fight against dominant culture bias. Nevertheless, due to other limitations such as lack of funding, difficulties in access, or just because those populations are themselves small; studying them may potentially translate into unavoidable sample size problems.

This is the case of our work, which has a too small sample size to do a more rigorous statistical treatment on the MST difference between high/low nutrition intake groups. An issue was highlighted by an anonymous reviewer during the peer review process. In particular, it would

be desirable to randomly construct k subnetworks of sizes n, m (with n<m); and then compare pairs of subnetworks of n nodes vs. subnetworks of m nodes; to statistically respond: How the difference of weight of the MST would be if randomly divide the data into two groups?

Although the sample size is not large enough to perform this gedankenexperiment we offer a theoretical argument.

Let's first consider that If there were no underlying ecological (external to the individual) population process, the mutual information matrix (MI) would be very homogeneous. These coefficients of the MI matrix are used as the weight of the networks, so weights are also very homogeneous. It would be expected then, that the value of the MST would change mainly in the function of the size of the network. This would mean that the MST of networks of m nodes would be greater than the MST of n nodes.

In our case study, the subnetwork with the highest number of nodes corresponds to the lowest intake of protein/lipids. However, the MST of the highest protein/fluid intake is the largest, in the other direction. Therefore, this result should not be attributed to chance.

In conclusion, we exemplify how network theory, through the MST metrics, has allowed us to unify and evaluate the connectivity of two biological systems from different substrates, neurons and GM, and assess how they are being affected by the same ecological pressure, the diet. The Brain and GM are two closely related systems that mature in parallel and whose level of communication will substantially impact the host's health in adulthood [52].

## Supporting information

**S1 File.**
(PDF)

**S2 File.**
(PDF)

**S3 File.**
(PDF)

**S4 File.**
(PDF)

## Acknowledgments

We are grateful to Margarita Muciño, Julio Naranjo from Xuajin Me'Phaa a non-governmental association, for their help in the liaison with the Me'Phaa community, their help in sample collection and logistics. We also are grateful to Santiago Martinez-Correa for their contribution in DNA amplification. Special thanks to Gregorio Iraola from institute Pasteur of Montevideo, Uruguay for his valuable bio-informatics orientation.

We would like to acknowledge the valuable assistance of Aida Elizondo-García in the fieldwork and data collection.

IG-S expresses his gratitude to the National Autonomous University of Mexico's PASPA-DGAPA program for its financial support during his sabbatical stay, which was instrumental in the development of this manuscript.

## Author Contributions

**Conceptualization:** Isaac G-Santoyo, Olga A. Rojas-Ramos, Luisa I. Falcón, Osiris Gaona, Daniel Cerqueda-García, Javier Nieto.

**Data curation:** Elvia Ramírez-Carrillo, Luisa I. Falcón, Osiris Gaona, Rosa María de la Fuente Rodríguez, Ariatna Hernández Castillo, Daniel Cerqueda-García, Andrés Sánchez-Quinto, Diego Hernández-Muciño.

**Formal analysis:** Elvia Ramírez-Carrillo, Oliver López-Corona.

**Funding acquisition:** Isaac G-Santoyo, Olga A. Rojas-Ramos, Luisa I. Falcón, Osiris Gaona, Javier Nieto.

**Investigation:** Elvia Ramírez-Carrillo, Oliver López-Corona, Olga A. Rojas-Ramos, Rosa María de la Fuente Rodríguez, Ariatna Hernández Castillo, Diego Hernández-Muciño.

**Methodology:** Elvia Ramírez-Carrillo, Isaac G-Santoyo, Oliver López-Corona, Olga A. Rojas-Ramos, Luisa I. Falcón, Osiris Gaona, Rosa María de la Fuente Rodríguez, Ariatna Hernández Castillo, Daniel Cerqueda-García, Andrés Sánchez-Quinto, Diego Hernández-Muciño, Javier Nieto.

**Project administration:** Isaac G-Santoyo.

**Resources:** Isaac G-Santoyo, Olga A. Rojas-Ramos, Luisa I. Falcón, Daniel Cerqueda-García, Andrés Sánchez-Quinto, Diego Hernández-Muciño, Javier Nieto.

**Supervision:** Isaac G-Santoyo, Javier Nieto.

**Visualization:** Elvia Ramírez-Carrillo, Oliver López-Corona.

**Writing – original draft:** Elvia Ramírez-Carrillo, Isaac G-Santoyo, Oliver López-Corona, Olga A. Rojas-Ramos, Luisa I. Falcón, Osiris Gaona, Rosa María de la Fuente Rodríguez, Ariatna Hernández Castillo, Daniel Cerqueda-García, Andrés Sánchez-Quinto, Diego Hernández-Muciño, Javier Nieto.

**Writing – review & editing:** Elvia Ramírez-Carrillo, Isaac G-Santoyo, Oliver López-Corona, Luisa I. Falcón, Osiris Gaona.

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
