## [Decision Letter · Decision Letter 0]

14 Sep 2022

PONE-D-22-17440Similar connectivity of gut microbiota and brain activity networks is mediated by animal protein and lipid intake in children from a Mexican indigenous population.PLOS ONE

Dear Dr. Ramírez-Carrillo,

Thank you for submitting your manuscript to PLOS ONE. After careful consideration, we feel that it has merit but does not fully meet PLOS ONE’s publication criteria as it currently stands. Therefore, we invite you to submit a revised version of the manuscript that addresses the points raised during the review process. The manuscript is interesting and presents a novel approach to shed light on the connectivity between the brain and gut microbiota networks. However, both reviewers highlights some flaws that need to be taken into account. Particularly, Rev. 1 indicates concerns about the experimental design, data handling, and statistical analyses. In addition, attention must be paid to several details (example: page 4, 33 children are mentioned, divided into 12 males and 19 females. But 19 + 12 = 31) and the English must be improved (see below). Finally, after responding to the queries of both reviewers and revising the manuscript, please resubmit it as a regular Word file with LINE NUMBERS, without any strange formatting to resemble a published PLOS article.

We look forward to receiving your revised manuscript.

Kind regards,

Nicoletta Righini, PhD

Academic Editor

PLOS ONE

Journal Requirements:

Please also provide as a Supporting Information file the questionnaire used in this study. If the questionnaire is in a language other than English, please provide as an additional Supporting Information file a version of the questionnaire translated into English.

Additionally, please note that PLOS ONE has specific guidelines on code sharing for submissions in which author-generated code underpins the findings in the manuscript. In these cases, all author-generated code must be made available without restrictions upon publication of the work. Please review our guidelines at https://journals.plos.org/plosone/s/materials-and-software-sharing#loc-sharing-code and ensure that your code is shared in a way that follows best practice and facilitates reproducibility and reuse

4. Please update your submission to use the PLOS LaTeX template. The template and more information on our requirements for LaTeX submissions can be found at http://journals.plos.org/plosone/s/latex

"We are grateful to Margarita Muciño, Julio Naranjo from Xuajin Me’Phaa a

non-governmental association, for their help in the liaison with the Me’Phaa community, their help in sample collection and logistics. We also are grateful to Santiago Martinez-Correa for their contribution in DNA amplification. IG-S explicitly acknowledges the UNAM and PASPA-DGAPA

program for the financial support granted to carry out his sabbatical stay and the development of this project"

"This project was funded by UNAM-PAPIIT [grant numbers IA209416, IA207019], DGAPA postdoctoral Fellowship, CONACYT Ciencia Básica [grant number 241744] and Instituto de Ecología-UNAM [L.I.F]."

6. Thank you for submitting the above manuscript to PLOS ONE. During our internal evaluation of the manuscript, we found significant text overlap between your submission and the following previously published work of which you are an author.

- https://www.biorxiv.org/content/10.1101/2020.07.25.221408v1

Please revise the manuscript to rephrase the duplicated text, cite your sources, and provide details as to how the current manuscript advances on previous work. Please note that further consideration is dependent on the submission of a manuscript that addresses these concerns about the overlap in text with published work.

**Additional Editor Comments (if provided):**

Minor details:

1) Must be "protein and lipid intake" (singular), not protein**s** and lipid**s** intake. Please change it throughout the manuscript.

2) Be consistent with how protein and lipid are written. It must be lower case, but sometimes they are written as Protein and Lipid (example, page 6)

3) Page 4: "and meanwhile animal-based products are available almost by hunting, it is consumed during special occasions and is not part of the daily diet".. Must be THEY ARE consumed (the subject is animal-based products).

4) GM is "gut microbiota". Why call them "gut bacteria microbiota"? (See Abstract). Bacteria are part of the microbiota, no need of repeating "bacteria". Moreover, it could be gut microbiota bacteria, not gut bacteria microbiota (which makes no sense). Also, bacteria are plural, so it's bacteria ARE/WERE, not bacteria IS/WAS (see page 5).

5) Page 13: Must be: This kind of population differS (third person singular)

6) Page 14: "but they are also present notably differences in factors...". Please rephrase. The sentence is not correct. Perhaps THERE ARE also differences? 

Reviewers' comments:

Reviewer's Responses to Questions

**Comments to the Author**

1. Is the manuscript technically sound, and do the data support the conclusions?

Reviewer #1: Partly

Reviewer #2: Yes

2. Has the statistical analysis been performed appropriately and rigorously? 

Reviewer #1: No

Reviewer #2: Yes

3. Have the authors made all data underlying the findings in their manuscript fully available?

Reviewer #1: Yes

Reviewer #2: Yes

4. Is the manuscript presented in an intelligible fashion and written in standard English?

Reviewer #1: Yes

Reviewer #2: Yes

5. Review Comments to the Author

**Reviewer #1:** Ramírez-Carrillo and collaborators presented the weight of Minimum Spanning Tree (MST) as a new metric to unify complex biological systems. And showed its application to brain and gut microbiota study in different diet intake groups.

The results are interesting especially the MST showed difference in gut microbiota while the traditional method for microbiota study did have significant difference. However, there was no statistic test for compare weight of MST and I also have some concern about the experiment design.

Major comment:

1. The author compared weight of the MST between high/low protein or lipid intake, showed the difference in percentage. But there is no statistic test to prove this difference is due to the high/low protein or lipid intake. In other words, how the difference of weight of the MST would be, if randomly divide the data in two groups?

2. The strategy to divide the children into two group based on mean proteins/lipids intake is not a proper way. For instance, for a normal distribution data, the data points are centered around mean. Arbitrarily assign those data to high or low won’t change the fact that they are close; for a u-shaped distribution, the major data points are separated, and far from mean, it’s proper to separate them by mean. Back to this study, it seems for protein intake, there are two peaks in distribution; but for lipid intake, the high and low are very close. It would be better to separate the lipid intake into three groups divided by the percentiles.

3. For the GM network, the author first selected the ASVs based on random forest to classify high/low intake group. I think this step introduced bias when build GM network. This network may not represent the full gut microbiota (only 700+ out of 1900+ ASVs are selected). I think the better way is to build a network based on all ASVs or the high abundant/prevalent ASVs to get the full picture of the gut microbiota.

4. The method section of “Network analysis for EEG and GM data” is not clear. a, the protein/lipid intake group part is belonged to “Protein and Lipids intake estimation”. b, stepwise details are needed for building brain power network and gut microbiota network. c, the introduction of MST should be in the beginning or in the end, not in between of brain power network and gut microbiota network, and the pseudo code is not necessary. d, a flowchart may help understand the steps.

Minor comment:

1. The format of citation is not consistent along the manuscript. Most of the citations are in parentheses, but a few are in superscript. like reference 20 in page 3, 33 in page 5, 42 in page 6, 54, 55 in page 13.

2. Page 4, study site and participants, the number of children is 33, but “12 males and 19 females”.

3. What is the correlation between protein and lipid intake?

4. The citation for R Package is not a link of pdf file. Please use citation(“package name”) in R to get the correct citations. Page 6, 7.

5. In Figure 3, difference percentage in brain power networks is using “low” group as reference. But in figure 5, the difference is calculated on “high” as reference. Suggest using one group as reference in the two figures.

6. Have you rarefied the ASV table for gut microbiota data?

7. Page 7, “With these families identified, using the dataset of ASVs (i.e. bacteria species) relative abundances in fecal samples, we used the Cooccur package (https://cran.r-project.org/web/packages/cooccur/cooccur.pdf) in R74 to construct a co-occurrence matrix and use it as weight in microbiota network”. It’s ASVs you identified, not “families”. ASVs are not “bacteria species”. And what is “R74”?

8. Page 12, “no Western diet”, should be non-western diet.

**Reviewer #2:** The manuscript “Similar connectivity of gut microbiota and brain activity networks is mediated by animal protein and lipid intake in children from a Mexican indigenous population” by Ramírez-Carrillo et al., is an original and interesting work in the area. It addresses the relationship between diet, microbiota and the brain, generating evidence to better understand this multidirectional relationship. In general, it is well written in a clear and fluent language. I consider it suitable for publication, after making MINOR REVISION. I list below mi suggestions:

1.Introduction:

• Page 3 of 16: Correct the format of citation number 20 in the text.

2. Materials and Methods:

• Page 5 of 16: In the “Gut microbiota composition” section, correct to 16S rRNA instead of rDNA.

• Page 6 of 16: In the “Network analysis for EEG and GM data” section, put spaces after the “=” sign and adjust throughout the document.

• Page 6 of 16: Remove “the” in (the 31 children).

• In methods, mentioned 33 participants and in results 31 children.

• Page 6 of 16: Figure 1. The figure must have a title and then, the explanation that is included. Please indicate the sample size as n = 9, n = 22, n = 11 and n = 20 and not just the isolated value.

• Page 6 of 16: Correct the format of citation number 42 in the text.

3. Results:

• Page 7of 16: Change in the text “study samples” to “study participants”. Also adjust the title of Table 1 in the same way.

• Page 7 of 16: Simplify the format of Table 1, placing the value and its deviation for males in a single column and in another for females (not in separate cells). Indicate in the header if it is mean or median and if it is standard error or standard deviation. Indicate in the title of the table that it is “according to sex”. It is recommended to add a column with the value of p and in the table footer indicate the realized statistical analysis (for example, t-test for independent samples).

• Page 8 of 16: Table 2. It must have a title and table footer. Unify format, avoid writing in capital letters, indicate if it is mean or median and if it is standard error or standard deviation.

• Page 9 of 16: Figure 2. Please include a figure title before the explanation.

• Page 11 of 16: Figure 4, unify 4a and 4b in lowercase as mentioned in the text, instead of A and B. It is recommended to cut the x-axis, for example up to 1000.

Page 11 of 16: “Finally, anthropometric and dietary patterns were not different between sex or age (Table 1). Hence, we did not include any of these variables as predictors of brain cortex or GM connectivity.” The authors do not present a statistical analysis showing that there are no differences. It is recommended to include it as part of the table.

4. Discussion:

• Page 13 of 16: In paragraph 2: “and in which essential factors,as diet, might profoundly influence the gut-brain signaling in adulthood (4).”, include a space and the word “such” before “as diet ”.

• Page 13 of 16: Correct the format of citations number 54 and 55 in the text.

5. Informed Consent Statement:

• All participants are between 5-10 years old. Please adjust the Informed Consent Statement, removing the adult participants.

6. PLOS authors have the option to publish the peer review history of their article (what does this mean?). If published, this will include your full peer review and any attached files.

Reviewer #1: No

Reviewer #2: **Yes: **Maria Esther Mejia-Leon

---

## [Author Response · Author response to Decision Letter 0]

16 Nov 2022

We are very thankful to editor and reviewers comments, we have follow them as better as we could and we are certain that they have improved the manuscript. Please find point to point response in the response letter file in table format.

---

## [Decision Letter · Decision Letter 1]

3 Jan 2023

PONE-D-22-17440R1Similar connectivity of gut microbiota and brain activity networks is mediated by animal protein and lipid intake in children from a Mexican indigenous population.PLOS ONE

Dear Dr. G-Santoyo,

Thank you for submitting your manuscript to PLOS ONE. After careful consideration, we feel that it has merit but does not fully meet PLOS ONE’s publication criteria as it currently stands. Therefore, we invite you to submit a revised version of the manuscript that addresses the points raised during the review process. The authors did a good job revising the manuscript according to the reviewers' comments. A few issues still need to be addressed.

We look forward to receiving your revised manuscript.

Kind regards,

Nicoletta Righini, PhD

Academic Editor

PLOS ONE

Journal Requirements:

Reviewers' comments:

Reviewer's Responses to Questions

**Comments to the Author**

1. If the authors have adequately addressed your comments raised in a previous round of review and you feel that this manuscript is now acceptable for publication, you may indicate that here to bypass the “Comments to the Author” section, enter your conflict of interest statement in the “Confidential to Editor” section, and submit your "Accept" recommendation.

Reviewer #1: All comments have been addressed

2. Is the manuscript technically sound, and do the data support the conclusions?

Reviewer #1: Yes

3. Has the statistical analysis been performed appropriately and rigorously? 

Reviewer #1: Yes

4. Have the authors made all data underlying the findings in their manuscript fully available?

Reviewer #1: Yes

5. Is the manuscript presented in an intelligible fashion and written in standard English?

Reviewer #1: Yes

6. Review Comments to the Author

**Reviewer #1: **

I think the authors answered most of my previous questions, although it takes a lot of time to check out every point since they didn’t mention the exact location in their revised manuscript to each question.

1. I still have the question regarding Lipid intake and protein intake. How many subjects belong to high lipid intake AND high protein intake? How many subjects belong to low lipid intake AND low protein intake? I think it’s important to know the proportion of overlapped subjects in high lipid and high protein group or low lipid and low protein group.

2. In their response they explained that their dataset is small and is not possible to have some statistic test on the MST difference between high/low nutrition intake group. I think it’s very important that they should also mention this limitation in their manuscript.

3. The author claimed they have corrected the format for R package citation, but they didn’t! line 241 and 249 are still two hyperlinks. I found the instruction for citation of these two packages below:

cooccur citation info (r-project.org), randomForest citation info (r-project.org).

What they need is just use the BibTeX entry from those links as the correct reference.

4. They now merged mean decrease gini of protein and lipid together in fig 5. That figure maybe misleading because in x axis the ASVs are arranges as the gini importance value decrease. When merge the two panels, it gives the reader the impression that the gini importance value ranks the same for the ASVs in protein intake and lipid intake. But I don’t think that’s true.

7. PLOS authors have the option to publish the peer review history of their article (what does this mean?). If published, this will include your full peer review and any attached files.

Reviewer #1: No

---

## [Author Response · Author response to Decision Letter 1]

11 Jan 2023

Reviewer #1

I think the authors answered most of my previous questions, although it takes a lot of time to check out every point since they didn’t mention the exact location in their revised manuscript to each question.

We apologize for this and the subsequent inconvenience, we are now mentioning the exact location of all changes, in green.

 I still have the question regarding Lipid intake and protein intake. How many subjects belong to high lipid intake AND high protein intake? How many subjects belong to low lipid intake AND low protein intake? I think it’s important to know the proportion of overlapped subjects in high lipid and high protein group or low lipid and low protein group.

There are 11 individuals with high protein intake, 9 with high lipid intake, and 6 individuals that are both in the high protein and lipid intake, corresponding to 42.8% shared. On the other hand, there are 20 individuals with low protein intake, 22 with low protein intake, and 16 both with low protein and lipid intake, corresponding to 61.5% shared. 

We included this test in lines 160-164

 In their response they explained that their dataset is small and is not possible to have some statistic test on the MST difference between high/low nutrition intake group. I think it’s very important that they should also mention this limitation in their manuscript.

We agree with reviewer, and have clearly state it at the end of the discussion section (lines 386-412): 

As we have discussed above, current research in the field is pointing to the importance of doing microbiota studies in nonwestern populations, and how minorities are usually under-represented in mainstream literature. This is relevant not just because they represent very interesting contrasting populations, but also to fight against dominant culture bias. Nevertheless, due to other limitations such as lack of funding, difficulties in access, or just because those populations are themselves small; studying them may potentially translate into unavoidable sample size problems. 

This is the case of our work, which has a too small sample size to do a more rigorous statistical treatment on the MST difference between high/low nutrition intake groups. An issue was highlighted by an anonymous reviewer during the peer review process. In particular, it would be desirable to randomly construct k subnets of sizes n, m (with n<m); and then compare pairs of subnets of n nodes vs. subnets of m nodes; to statistically respond: How the difference of weight of the MST would be if randomly divide the data into two groups?

Although the sample size is not large enough to perform this gedankenexperiment we offer a theoretical argument. 

Let's first consider that If there were no underlying ecological (external to the individual) population process, the mutual information matrix (MI) would be very homogeneous. These coefficients of the MI matrix are used as the weight of the networks, so weights are also very homogeneous. It would be expected then, that the value of the MST would change mainly in the function of the size of the network. This would mean that the MST of networks of m nodes would be greater than the MST of n nodes. 

In our case study, the subnetwork with the highest number of nodes corresponds to the lowest intake of protein/lipids. However, the MST of the highest protein/fluid intake is the largest, in the other direction. Therefore, this result should not be attributed to chance.

The author claimed they have corrected the format for R package citation, but they didn’t! line 241 and 249 are still two hyperlinks. I found the instruction for citation of these two packages below:

cooccur citation info (r-project.org), randomForest citation info (r-project.org).

What they need is just use the BibTeX entry from those links as the correct reference.

Sorry for this, we have now taken care of it and now it is found on lines 244 and 250-251 as following:

 “...implemented in R under the randomForest package [50], and we identified essentia” 

“... for this reason, we used the complete ASVs data set as input to a co-occurrence analysis using Cooccur package in R [51].”

Also corrected citation on line 226:

“Mutual Information (MI) matrix using the ‘mpmi’ R Package [46].”

 They now merged mean decrease gini of protein and lipid together in fig 5. That figure maybe misleading because in x axis the ASVs are arranges as the gini importance value decrease. When merge the two panels, it gives the reader the impression that the gini importance value ranks the same for the ASVs in protein intake and lipid intake. But I don’t think that’s true.

Thank you for this detailed review. We really appreciate all your work. We modified the figure to take care of this comment by separating into two panels

---

## [Editor Report · Decision Letter 2]

23 Jan 2023

Similar connectivity of gut microbiota and brain activity networks is mediated by animal protein and lipid intake in children from a Mexican indigenous population.

PONE-D-22-17440R2

Dear Dr. G-Santoyo,

We’re pleased to inform you that your manuscript has been judged scientifically suitable for publication and will be formally accepted for publication once it meets all outstanding technical requirements.

Kind regards,

Nicoletta Righini, PhD

Academic Editor

PLOS ONE
---

## [Editor Report · Acceptance letter]

27 Jan 2023

PONE-D-22-17440R2 

Similar connectivity of gut microbiota and brain activity networks is mediated by animal protein and lipid intake in children from a Mexican indigenous population. 

Dear Dr. G-Santoyo:

I'm pleased to inform you that your manuscript has been deemed suitable for publication in PLOS ONE. Congratulations! Your manuscript is now with our production department. 

Kind regards, 

on behalf of

Dr. Nicoletta Righini 

Academic Editor

PLOS ONE